# Evolution of the nutritional status and factors associated with undernutrition in children under five of age between 2014 and 2017 in 24 health districts of Burkina Faso

T. Bernadette Picbougoum[1,2]*, M.A. Serge Somda[1,3], Isidore T. Traoré[1,4], Julia Lohmann[5], Manuela De Allegri[5], Hervé Hien[1,6], Nicolas Méda[7], Annie Robert[2]

**1** Institut National de Santé Publique (INSP)/ Centre MURAZ, Bobo-Dioulasso, Burkina Faso, **2** Institut de recherche expérimentale et clinique, Pôle Epidémiologie et Biostatistique, Université catholique de Louvain (UCLouvain), Brussels, Belgium, **3** Unité de Formation et de Recherche en Sciences exactes et appliquées, Université Nazi Boni, Bobo-Dioulasso, Burkina Faso, **4** Institut Supérieur des Sciences de la Santé, Université Nazi Boni, Bobo-Dioulasso, Burkina Faso, **5** Heidelberg Institute of Global Health, University Hospital and Medical Faculty, Heidelberg University, Heidelberg, Germany, **6** Institut de Recherche en Sciences de la Santé (IRSS), Direction Régionale de l'Ouest, Bobo-Dioulasso, Burkina Faso, **7** Unité de Formation et de Recherche en Sciences de la Santé (UFR/SDS), Université Joseph Ki-Zerbo, Ouagadougou, Burkina Faso.

\* bernadette.picbougoum@centre-muraz.bf, tarwendpanga.picbougoum@uclouvain.be

## Abstract

The prevalences of undernutrition in children under five years of age appeared to decrease over the last decade in Burkina Faso. However, the country is now facing new health and security challenges that could threaten this progress. Therefore, it is essential to monitor the evolution of the situation within this specific context. We aimed to assess trends in undernutrition among children under 5 years of age, as well as the factors associated with it, between 2014 and 2017 in Burkina Faso. A study based on secondary analysis of the data from 2013 and 2017 surveys, conducted in 24 districts across six health regions, was carried out. We worked with the household databases to select two sub-samples: 9,259 children from 2014 and 12,119 children from 2017. We assessed anthropometric indicators using World Health Organization standards, analyzed their proportions between 2017 and 2014, and according to the health regions and children's age groups. We examined the association of stunting and underweight with children, mothers, and households' characteristics in 2017 and for two selected health regions, using logistic regression. From 2014 to 2017, the proportion of severe stunting and severe underweight decreased respectively from 24.8% to 7.9% (*p < 0.001*) and from 19.6% to 8.8% (*p < 0.001*) certainly due to nutrition and health initiatives. As in 2014, stunting was associated with sex, living in poorest household with AdjOR = 1.31 (95%CI: 1.14 -1.50), or in household having three and more children under five years with AdjOR = 1.28 (95%CI: 1.12 -1.47). This study demonstrated a period of significant progress

**Data availability statement:** All data are available in the paper and Database is accessible, attached in Supporting Information files.

**Funding:** The author(s) received no specific funding for this work.

**Competing interests:** The authors have declared that no competing interests exist.

in undernutrition in children. But it identified the persistence of associated factors contributing to the high prevalence of undernutrition in Burkina Faso, where insecurity has become a prevailing issue. Further studies are needed to assess the sustainability of encouraging progress in children's nutritional status.

## Introduction

Malnutrition includes both undernutrition and overnutrition. Undernutrition, resulting from inadequate intake or assimilation of nutrients necessary to promote growth and prevent chronic or acute diseases, is often characterized by stunting, wasting, underweight, and micronutrient deficiencies [1,2]. Malnutrition in children under five years of age remains a significant global issue, despite considerable progress in improving both the quality and quantity of the world's food supply in recent years [3]. In 2017, an estimated 155 million children under five years of age were stunted, and 52 million were wasted [4]. Asia and Africa continue to bear the greatest burden of malnutrition, with more than half of all stunted children and two-thirds of all wasted children under five years of age living in Asia, and over one-third of stunted children and a quarter of wasted children living in Africa [5]. After remaining relatively unchanged since 2015, the prevalence of undernourishment continued to rise before the COVID-19 pandemic and consequently to more crisis [6].

Burkina Faso is a country facing two major challenges: on the health front, with the continuing endemicity of malaria and the dengue epidemic, and on the security front, with a worsening insecurity situation that has been escalating for more than seven years. This security situation began in 2016 with terrorist attacks, which have intensified and spread over time. It has resulted in internal population movements from their villages to others, and the closure or minimal operation of certain health facilities, leading to reduced access to health and nutrition services [7]. In this country, a global decrease in the prevalence of undernutrition was observed between 2009 and 2017 [8,9]. For instance, stunting decreased from 35.1% in 2009 to 29.1% in 2014 and 21.2% in 2017; underweight decreased from 26% in 2009 to 20.1% in 2014 and 16.2% in 2017; and wasting decreased from 11.3% in 2009 to 8.6% in 2014 and 8.6% in 2017 [8–10]. However, prevalences remain high, although the situation has improved in recent years. In addition, this evolution often hides disparities between regions and between years. For example, the highest regional prevalences of stunting were observed over the 10-year period from 2010 to 2017 in the Est (39.6% and 34.5%), Cascades (from 38.0% to 30.5%), and the Sahel (from 36.5% to 38.9%) [9,11]. On top of that, several factors have been identified in previous studies in Burkina Faso as the main contributors to undernutrition in children under five years of age [12–17]. In this way, undernutrition continues to be a burden despite the several efforts made in the fight, including nutrition-specific and health-sensitive interventions that improve health and should influence the nutritional status of children. It affects children's health, contributes to the impoverishment of households, affects the long-term development of country, and presents challenges related to access

to healthcare and nutrition. National policies, programs, and strategies to combat malnutrition, as well as those aimed at improving the functioning of the healthcare system, have been implemented over the last decade. Examples include the integrated management of childhood illnesses, the integrated management of wasting, vitamin A supplementation, infant and young child feeding practices, performance-based financing, and the national nutrition program [18]. Moreover, the spread of two major challenges in health and security, with serious consequences, added to existing individual, community, and environmental factors, is likely to hinder the favorable evolution of children's nutritional status. However, there has been very little analysis of trends in children's nutritional indicators over the last ten years, considering two phases: before and during these social and health phenomena, as well as the factors associated with the persistence of undernutrition in some health regions. Therefore, it is necessary to monitor the evolution of the situation within this specific context. This will allow us to describe the country's status regarding children's nutritional health, assess the impact of the various actions taken to improve people's health, and evaluate the trend of associated factors. Our study aims to assess trends in undernutrition among children under 5 years of age, as well as the factors associated with it, between 2014 and 2017 in Burkina Faso.

## Materials and methods

### Ethics statement

This study obtained clearance from the authors' research institutions' authorities. It was conducted in accordance with the guidelines outlined in the Declaration of Helsinki, and all procedures involving research study participants were approved by the Ethical Committee of Heidelberg University (S-272/2013) and the National Ethics Committee of Burkina Faso (N° 2013-7-066). Written informed consent was obtained from all study participants and parents/guardians of the children and others under 18 years of age before inclusion. Individual data from the study participants were anonymized prior to analysis.

### Study design

This study was based on a secondary analysis of data from two surveys [19]. In 2013 and 2017, two surveys were conducted in health facilities and households in 24 districts across six health regions of the country. Details of the study design are reported elsewhere [20–22]. We worked with the databases of those households to select two sub-samples of children for our study. The first household survey was carried out from October 15, 2013, to March 15, 2014, to collect individual and household socio-demographic characteristics and health indicators for child health, including reported health and nutritional status. The second survey was conducted from April 10, 2017, to July 23, 2017. The study included villages in six health regions of Burkina Faso: Boucle du Mouhoun, Centre-Nord, Centre-Ouest, Nord, Sud-Ouest, and Centre-Est. Twenty-four districts (Nouna, Solenzo, Kongoussi, Kaya, Koudougou, Sapouy, Gourcy, Ouahigouya, Batié, Diébougou, Ouargaye, Tenkodogo, Boromo, Toma, Barsalogho, Ziniaré, Nanoro, Réo, Yako, Boussé, Dano, Gaoua, Zabré, and Manga. The surveys were conducted among households in the areas of these health facilities. Household survey data provide pre- and post-implementation assessments of selected children's health and nutritional outcomes.

### Sampling and study participants

Both the 2014 and 2017 surveys collected the same set of primary data from the households. The households surveyed were selected based on the criteria of having at least one case of pregnancy or birth in the two years preceding the survey. The household surveys used a three-stage sampling method to select primary health care facilities in the health area, from which one village was randomly selected from each cluster.

Fifteen households were surveyed in each of the 537 enumeration sections (i.e., villages) within the health areas of 537 Centre de Santé et de Promotion Sociale (CSPSs) from 24 district projects, selected from the baseline survey. Overall,

7,898 households were investigated. When a baseline household was not found or did not meet the eligibility criteria, it was replaced by the neighboring household or the closest household meeting the criteria (households with at least one pregnant woman or woman who gave birth in the last two years). The surveys covered heads of households, pregnant women, women who had given birth in the last two years at the time of the study, and children under five years old, focusing on socio-demographic, economic, and health characteristics. Malaria infection and anemia were assessed using the Malaria Rapid Diagnostic Test (RDT) and Hemocue® test, respectively, in all children under five years of age, pregnant women, and women who had delivered in the last two years at the time of the study, provided they were present in the household during the visit. Anthropometric data, such as weight, height, and mid-upper arm circumference (MUAC), were also collected.

## Study variables

We used three indicators of nutritional status: stunting, wasting, and underweight in children under five years of age as outcome variables. These are indicators that measure the nutritional status of children. They are described in our previous manuscript [17] according to the Waterlow definitions and the World Health Organization (WHO) 2006 standards [23].

We included socio-demographic and economic variables in the analysis. These included the sex of the child (male or female), their age, place of residence, mother's age, mother's educational attainment, household economic level, and the number of children under five years old in the household. Mothers' educational attainment was classified into three levels: no schooling, primary school, and secondary or higher education. Household wealth was assessed using the standard multiple component analysis (MCA) method to create a wealth index based on the following assets, classifying households into quintiles based on housing (type of building, number of rooms, water, and energy supply), assets (TV, radio, fridge, etc.), and ownership of houses, fields, and animals. Finally, we classified households into the country's climatic zones (Sahelian, Sudano-Sahelian, and Sudanese).

## Analysis

The research question requires a description of a situation in space and time. Therefore, comparative descriptive analysis is the appropriate method for conducting this research. We performed a descriptive analysis of the economic and socio-demographic characteristics of households, children, and mothers in each sample. Pearson's chi-square test was applied to compare the sample characteristics. We then determined the levels of stunting, wasting, and underweight by calculating the proportion of children classified according to the corresponding z-score, using the total number of children as the denominator. The coexistence of these three forms of undernutrition in the same child was assessed by performing cross-analysis between stunting, wasting, and underweight. For the Height Z-score and Weight Z-score indicators, the mean Z-score was calculated to compare 2014 and 2017, according to the health regions or the age groups of the children. This choice is guided by the fact that stunting and underweight reflect chronic undernutrition, making them appropriate for assessing trends, while wasting reflects acute undernutrition. Afterward, we performed a comparison of means using ANOVA to assess the existence of significant differences in Z-scores between regions, children's age groups, and years. The prevalence of the two forms of undernutrition (stunting and underweight) was also compared between 2014 and 2017, as well as by health regions and children's age groups, through proportion comparison tests. Two regions where these proportions particularly increased were selected to assess the associated factors. Then, we used logistic regression (LR) models to assess the associations of stunting and underweight with the explanatory factors in 2017 for these two regions. For this, the dependent variables were expressed as dichotomous variables, as presented elsewhere [17]. We used a full model with all the explanatory factors and then analyzed each one using the LR test. Significant factors with a p-value < 0.05 were considered at the end of these multivariable models. Odds ratios (OR) were provided along with their 95% confidence intervals (95% CI). Afterward, we applied the Pearson or Hosmer-Lemeshow goodness-of-fit test to examine whether the models were correctly specified. We performed the analysis using Stata version 18.0 Standard Edition (StataCorp, Texas), Microsoft Excel 2010, and R version 4.3.2.

## Results

Our study is based on two samples of children aged 0–59 months: one from 2014, comprising 9,259 children, and another from 2017, comprising 12,119 children. Among them, 2,371 children who had been surveyed in 2014 were tracked down in 2017. This paired sample will be specifically analyzed in future studies.

In both samples, approximately 50% of the children were male, and 93% resided in rural areas. Approximately 7% and 6% of the children's mothers in the 2014 and 2017 samples, respectively, were teenagers (Table 1).

The proportion of children under five suffering from stunting decreased from 44.4% in 2014 to 24.9% in 2017, with severe cases dropping from 24.8% to 7.9%, representing a reduction of 16.9 percentage points. As for underweight, it decreased by 12.6 percentage points, with severe cases showing a reduction of 10.8 percentage points. For wasting, the proportion decreased from 28.5% in 2014 to 20.1% in 2017, with severe cases dropping from 17.1% to 9.3% (Table 2).

The proportion of children suffering from both stunting and wasting, as well as underweight, decreased from 10.2% to 4.8% between 2014 and 2017 (Fig 1).

Analysis of the mean Z-scores also showed a decrease in the Height Z-score, from -1.77 (SD = 2.32) in 2014 to -1.00 (SD = 1.90) in 2017. A similar downward trend was observed in the mean Weight Z-score, which decreased from -1.51 (SD = 1.83) in 2014 to -1.18 (SD = 1.54) in 2017.

According to the children's age groups, the proportion of stunting decreased between 2014 and 2017 in the 0–11 and 12–23 month age groups, while it increased in the 24–35, 36–47, and 48–59 month age groups (Table 3).

**Table 1. Sociodemographic and socioeconomic characteristics of children under five years and their mothers in 2014 and 2017.**

| | 2014 N = 9,259 n (%) | 2017 N = 12,119 n (%) | p-value | | 2014 N = 9,259 n (%) | 2017 N = 12,119 n (%) | p-value |
|---|---|---|---|---|---|---|---|
| **Sex** | | | *< 0.001* | **Mother's age (years)** | | | *< 0.001* |
| Male | 4,671 (50.5) | 6,065 (50.1) | | [15 - 19] | 647 (7.0) | 722 (6.0) | |
| Female | 4,588 (49.5) | 6,054 (50.0) | | [20 - 29] | 4,906 (53.0) | 5,899 (49.4) | |
| **Age (months)** | | | *< 0.001* | [30 - 39] | 3,130 (33.8) | 4,459 (37.3) | |
| [0 - 11] | 2,927 (31.6) | 3,305 (27.3) | | [40 - 49] | 574 (6.2) | 870 (7.3) | |
| [12 - 23] | 2,644 (28.6) | 2,751 (22.7) | | **Mother educational level** | | | *< 0.001* |
| [24 - 35] | 1,144 (12.4) | 1,540 (12.7) | | None | 9,209 (99.6) | 10,331 (85.4) | |
| [36 - 47] | 1,359 (14.7) | 2,542 (21.0) | | Primary school | 36 (0.4) | 1,290 (10.7) | |
| [48 - 59] | 1,185 (12.8) | 1,981 (16.4) | | Secondary & higher school | 5 (0.1) | 470 (3.9) | |
| **Residence** | | | *< 0.001* | **H Economic level*** | | | *< 0.001* |
| Urban | 653 (7.1) | 883 (7.3) | | 1st quintile (poorest) | 1,619 (17.5) | 2,137 (17.6) | |
| Rural | 8,606 (93.0) | 11,236 (92.7) | | 2nd quintile (poor) | 1,671 (18.1) | 2,336 (19.3) | |
| **Climatic regions** | | | *< 0.001* | 3rd quintile (middle) | 1,842 (19.9) | 2,388 (19.7) | |
| Sahelian | 3,216 (34.7) | 4,353 (35.9) | | 4th quintile (rich) | 2,017 (21.8) | 2,653 (21.9) | |
| Sudano-Sahelian | 5,348 (57.8) | 6,933 (57.2) | | 5th quintile (richest) | 2,110 (22.8) | 2,605 (21.5) | |
| Sudanese | 695 (7.5) | 833 (6.9) | | | | | |
| **Number of children U5** | | | | | | | |
| One | 2,788 (30.1) | 1,532 (12.6) | | | | | |
| Two | 4,522 (48.8) | 4,831 (39.9) | | | | | |
| Three and more | 1,949 (21.1) | 5,756 (47.5) | | | | | |

*According to the standard multiple component analysis (MCA) methods: housing (type of building, number of rooms, water, and energy supply), assets (TV, radio, fridge, etc.), houses and fields owned, and animals

**Table 2. Nutritional status of children under five years of age in Burkina Faso in 2014, 2017.**

| | 2014 N = 9,259 n (%) | 2017 N = 12,119 n (%) | p-value | Decrease (95% CI) |
|---|---|---|---|---|
| **Height Z-score** | | | **< 0.001** | |
| <- 3.0: Severe stunting | 2,292 (**24.8**) | 953 (**7.9**) | | **16.9** (15.9 - 17.9) |
| [-3.0; -2.0[: Moderate stunting | 1,815 (**19.6**) | 2,059 (**17.0**) | | **2.6** (1.6 - 3.7) |
| ≥ - 2.0: Normal | 5,152 (55.6) | 9,107 (75.2) | | – |
| **Weight-for-height Z-score** | | | *0.08* | |
| <- 3.0: Severe wasting | 1,584 (**17.1**) | 1,131 (**9.3**) | | **7.8** (6.8 - 8.7) |
| [-3.0; -2.0[: Moderate wasting | 1,051 (**11.4**) | 1,312 (**10.8**) | | **0.6** (-0.3 - 1.4) |
| [-2.0; +2.0]: Normal | 5,517 (59.6) | 9,315 (76.9) | | – |
| > +2.0: Overweight/ Obese | 1,107 (12.0) | 361 (3.0) | | – |
| **Weight Z-score** | | | **< 0.001** | |
| <- 3.0: Severe underweight | 1,813 (**19.6**) | 1,066 (**8.8**) | | **10.8** (9.8 - 11.7) |
| [-3.0; -2.0[: Moderate underweight | 1,694 (**18.3**) | 1,996 (**16.5**) | | **1.8** (0.8 - 2.9) |
| ≥ - 2.0: Normal | 5,752 (62.1) | 9,057 (74.7) | | – |

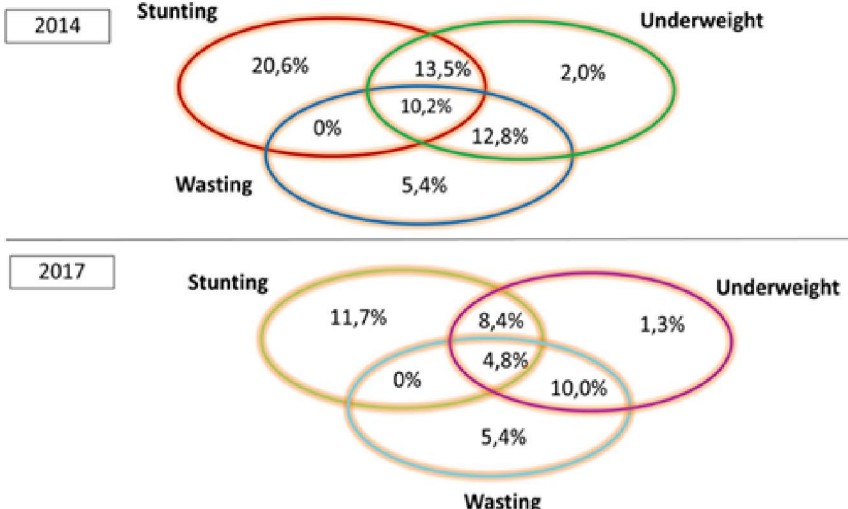

**Fig 1. Coexistence of the three forms of undernutrition among children under five in Burkina Faso, 2014 and 2017.**

**Table 3. Trends in the proportion of stunting and underweight among children by age group in Burkina Faso.**

| | Stunting (%) | | | Underweight (%) | | |
|---|---|---|---|---|---|---|
| Age (month) | 2014 | 2017 | Difference (95% CI) | 2014 | 2017 | Difference (95% CI) |
| [0 - 11] | 20.1 | 10.8 | **9.3** (7.7 - 11.0) | 23.9 | 21.9 | **2.0** (-0.01; 4.3) |
| [12 - 23] | 33.2 | 31.1 | **2.1** (-0.01 - 4.3) | 28.2 | 28.3 | **-0.1** (-2.3; 2.0) |
| [24 - 35] | 15.2 | 18.2 | **-3.0** (-4.8 - -1.3) | 14.1 | 14.8 | **-0.7** (-2.3; 1.1) |
| [36 - 47] | 17.2 | 23.8 | **-6.6** (-8.5 - -4.7) | 17.9 | 18.7 | **-0.8** (-2.6; 1.1) |
| [48 - 59] | 14.3 | 16.1 | **-1.8** (-3.5 - -0.1) | 15.9 | 16.4 | **-0.5** (-2.3; 1.3) |

The mean Height Z-scores and Weight Z-scores decreased between 2014 and 2017 across all age groups of children. Variance analysis (ANOVA) showed a significant difference in the mean Height Z-scores (p-values < 0.001) between at least two age groups in both 2014 and 2017. It showed a significant difference in the mean Weight Z-scores (p-values < 0.001) between at least two age groups in both 2014 and 2017 (S1 Fig).

However, regarding the evolution of proportions by region, we observed a trend toward stagnation or even a slight increase in stunting between 2014 and 2017 in each region, contrary to the overall data. The same observation was made for underweight, with a notable increase in the Centre-Est and Centre-Nord regions based on our data. However, the results of the nutritional surveys showed a significant increase in the proportion of stunting in the health regions of Centre-Nord and Nord (Table 4).

As for the mean Z-scores by health region, Centre-Ouest and Nord had mean Height Z-scores ≤ -1.64 in 2014, while all regions had means > -1.28 in 2017. Boucle du Mouhoun, Centre-Est, Sud-Est, and Nord had mean Weight Z-scores ≤ -1.64 in 2014, while all regions had means > -1.28 in 2017. Variance analysis (ANOVA) by region showed a significant difference in the mean Height Z-scores and Weight Z-scores between at least two regions in 2014 (p-values < 0.001). In 2017, there was no significant difference in the mean Height Z-scores and Weight Z-scores (p-values = 0.067 and 0.14, respectively; S2 Fig). The distribution of mean Height Z-scores and Weight Z-scores across the 24 districts was also analyzed, but no significant differences were observed between the means.

Logistic regression analysis revealed that the same factors influencing stunting in 2014 were also present in 2017, with an additional factor identified: the mother's educational level. In the North, stunting was associated with place of residence, mother's education level, and mother's occupation, while in the Centre-Nord, it was associated with the number of children under five years in the household. Underweight was associated with climatic regions in the Nord and Centre-Nord, the mother's level of education in the Nord, and the household economic level in the Centre-Nord (Figs 2, 3).

## Discussion

The importance of monitoring children's growth has been well established for decades [24]. Linear growth is one of the best overall indicators of children's well-being and serves as an accurate marker of inequalities in human development. This is tragically reflected in the millions of children worldwide who fail to achieve their linear growth potential due to sub-optimal health conditions, inadequate nutrition, and poor care. Moreover, they suffer severe and irreversible physical and cognitive damage associated with stunted growth. Stunting often goes unnoticed in communities where short stature is so prevalent that it is perceived as normal. The difficulty in visually identifying stunted children, combined with the lack of routine linear growth assessments in primary health care services, explains the delayed recognition of the magnitude of this hidden scourge. Stunting is now recognized as a major global health priority and the focus of several prominent initiatives [25,26].

Our study aimed to contribute to the existing body of scientific evidence on the trends in undernutrition in developing countries, with a focus on Burkina Faso. Undernutrition was assessed using the principal component analysis technique,

**Table 4. Trends in the proportion of stunting and underweight by health regions in Burkina Faso.**

| Region | Stunting (%) | | Underweight (%) | |
|---|---|---|---|---|
| | 2014 | 2017 | 2014 | 2017 |
| Boucle du Mouhoun | 19.4 | 19.1 | 17.4 | 17.7 |
| Centre-Est | 12.2 | 13.0 | 8.1 | 12.1 |
| **Centre-Nord** | **19.9** | **20.6** | **14.9** | **20.3** |
| Centre-Ouest | 18.6 | 17.6 | 26.7 | 18.6 |
| **Nord** | **23.0** | **23.9** | **27.9** | **25.7** |
| Sud-Ouest | 7.0 | 5.8 | 5.2 | 5.7 |

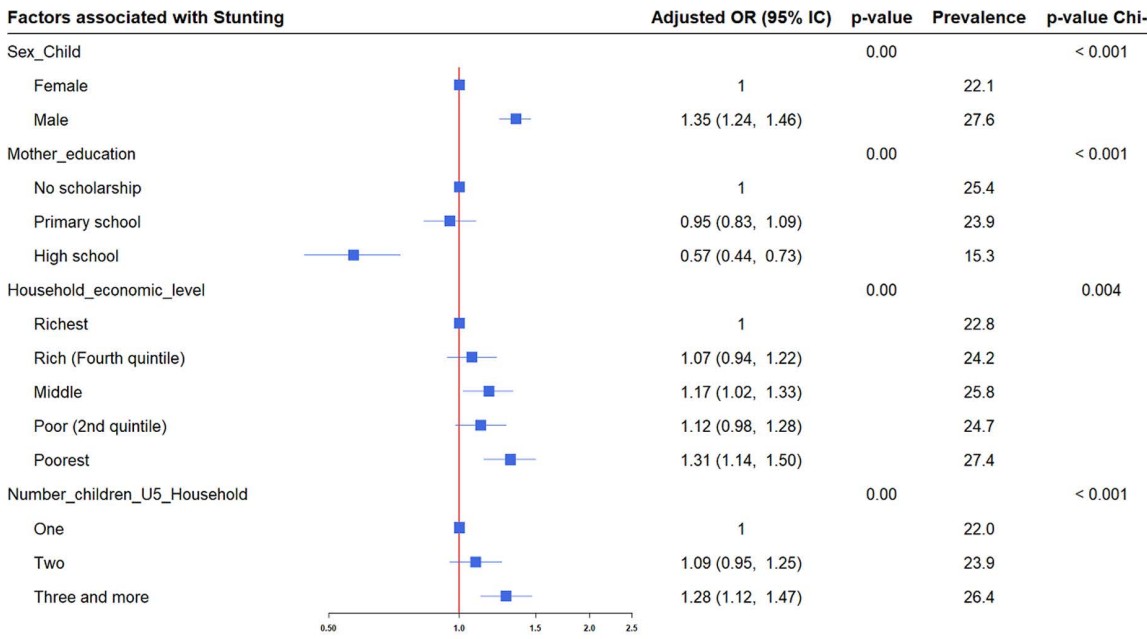

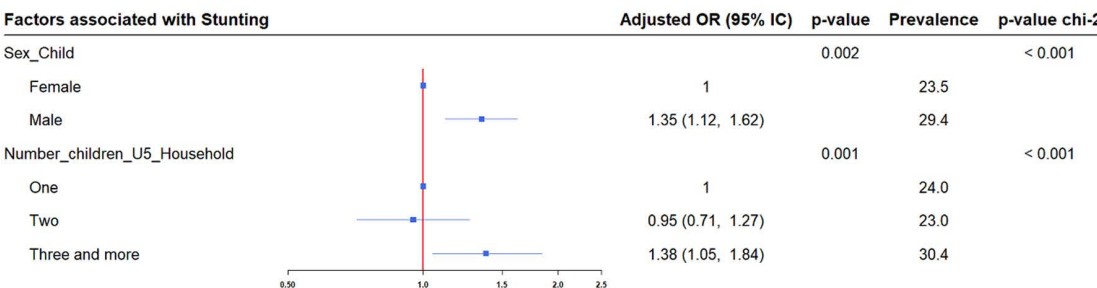

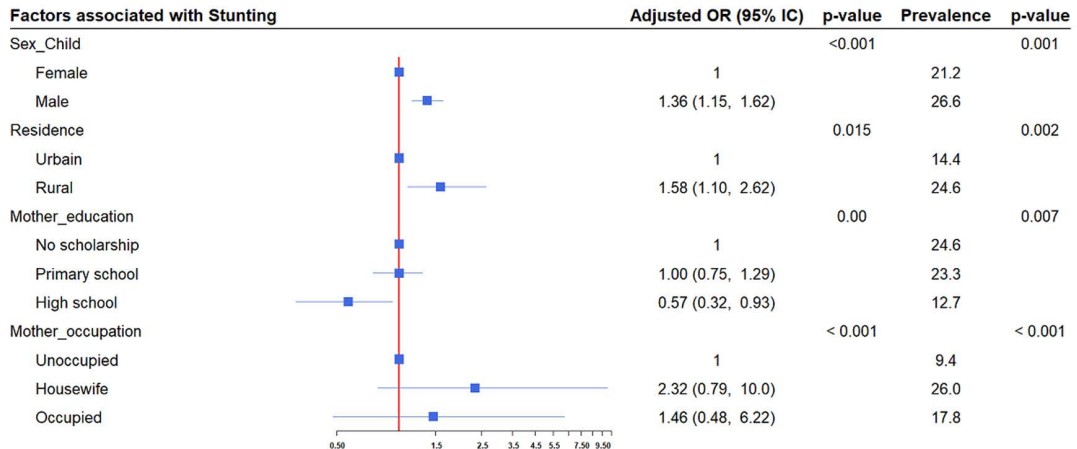

**Fig 2. Factors associated with stunting Among Children Under Five Years in 2017.** The analysis was conducted for the 24 health districts, as well as in the Centre-Nord and Nord regions, using multivariate logistic regression and Pearson's chi-square test. The red solid vertical line represents the reference line. Blue square markers indicate the adjusted odds ratios, while the horizontal blue lines represent the 95% confidence intervals.

PLOS Global Public Health

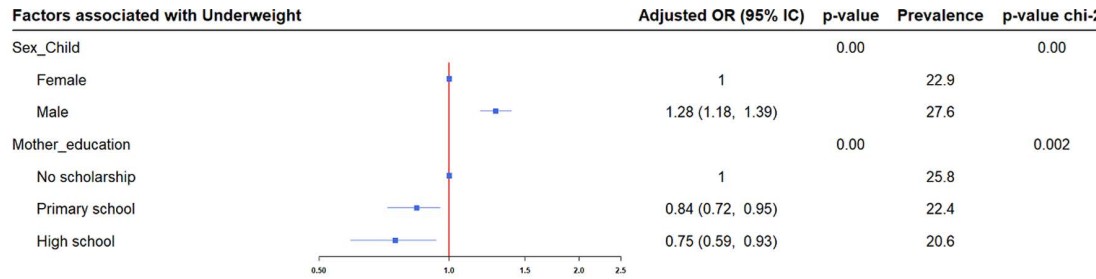

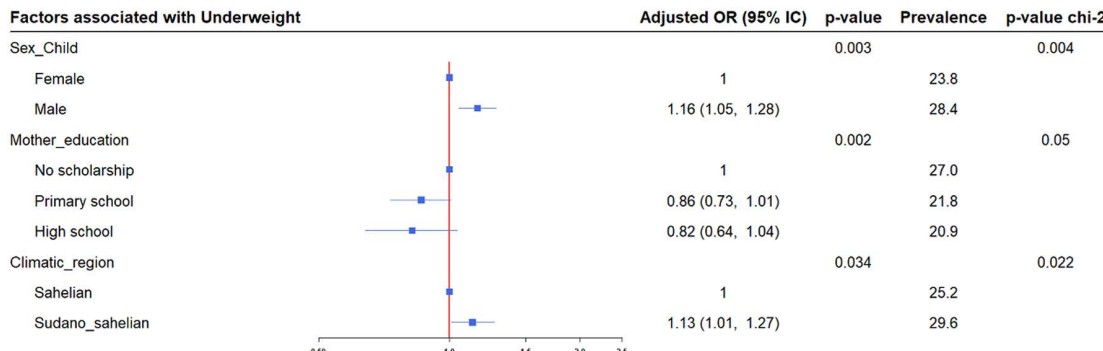

**Fig 3. Factors Associated with Underweight Among Children Under Five Years in 2017.** The analysis was conducted across the 24 health districts, as well as in the Centre-Nord and Nord regions, using multivariate logistic regression and Pearson's chi-square test. The red solid vertical line represents the reference line. Blue square markers indicate the adjusted odds ratios, and the horizontal blue lines represent the 95% confidence intervals.

derived from the three standard anthropometric indices. However, the in-depth analyses focused on two primary forms of undernutrition: stunting and underweight. The study assessed the trends in the nutritional status of children under five years of age and the factors associated with undernutrition between 2014 and 2017, based on surveys conducted for various primary purposes in Burkina Faso.

The study identified high levels of stunting, underweight, and wasting in 2014, with a significant downward trend observed in 2017, particularly for severe cases. The reduction in moderate cases was minimal across all three indicators. Severe cases undergoing treatment may transition to a moderate phase before achieving full recovery. The proportion of children affected by a combination of stunting, wasting, and being underweight has also decreased. Our study revealed a

significant reduction in the prevalence of undernutrition between 2014 and 2017. These indicators show an encouraging trend. This improvement could be attributed to better living conditions within the population. For instance, Burkina Faso's Human Development Index (HDI) improved significantly over the years, rising from 0.293 to 0.452 between 2000 and 2009, and from 0.405 to 0.439 between 2014 and 2017. Thus, this suggests an improvement in household living standards. Therefore, the healthcare system has improved through various health and nutritional interventions, particularly targeting children, women, and vulnerable groups. During this period, the socio-political context remained unchanged. This progress aligns with the objectives of the WHO, which adopted a resolution on maternal, infant, and young child nutrition in 2012. The resolution established six global targets to hold the world accountable for reducing malnutrition by 2025, including an annual 3.9% reduction in stunting and a goal to decrease the number of stunted children [25]. However, the levels of underweight, stunting, and wasting observed in this study were higher than the national averages for the same years. Nationally, stunting was recorded at 29.1% in 2014 and 21.2% in 2017; underweight at 20.1% in 2014 and 16.2% in 2017; and wasting remained steady at 8.6% in both 2014 and 2017 [9,10]. These differences may be attributed to the fact that our study focused on six health regions specifically selected for intervention due to their poor maternal and child health indicators, as reflected in our findings. However, the prevalences we observed align with most findings from sub-Saharan Africa. [1,2,13,27–37]. Moreover, this encouraging trend was observed during the onset of the country's insecurity crisis. As terrorist attacks have spread and intensified since 2017, their consequences have been significant on both health and socio-economic levels. This has led to the internal displacement of populations from their villages to other areas, as well as the closure or limited operation of certain health facilities, resulting in reduced access to health and nutrition services. In 2022, the number of internally displaced persons was estimated at 1,882,391 moved persons according to Conseil National de Secours d'Urgence et de Réhabilitation (CONASUR) [7]. Reports from national nutritional surveys (SMART) have already indicated an increased prevalence of undernutrition [38–40].

Based on children's age groups, a trend was observed where proportions either stagnated or increased between 2014 and 2017. Primarily among groups over 24 months old, the proportions of stunting were higher in 2017 compared to 2014. This difference was most pronounced in the 36–47 months age group for stunting. From 24 months onward, the proportions of children in these age groups were higher in 2017 than in 2014. This could be explained by the fact that, as the years go by, children move into higher age groups if they are not treated. Additionally, this result aligns with several studies that found an increase in the prevalence of stunting among children with birth intervals of less than 24 months [25,26,2,32,35,41,42].

In terms of the mean Z-score, as well as the Height and Weight Z-scores, there was a decline between 2014 and 2017. Furthermore, significant differences were found between the mean Height Z-scores and the mean Weight Z-scores across age groups.

In 2014, three out of six regions reached the 25th percentile (> -1.28) for Height Z-score, while in 2017, all six regions achieved this. For Weight Z-score, the number of regions with this threshold increased from none to all six. Moreover, variance analysis revealed a significant difference between the mean Height Z-scores and Weight Z-scores in at least two regions in 2014. In 2017, there was no significant difference between the mean Height Z-scores and Weight Z-scores across regions. These regional differences were observed in the national rankings and have been demonstrated in several studies. Regional factors, such as the adverse effects of weather on agricultural harvests, poor soil quality, low rainfall patterns, and others, contributed to poorer nutritional status in children within rural farming populations [13,43–45].

According to the associated factors, the trend does not show a decline, unlike the forms of undernutrition. In addition to the factors identified in 2014 [17], two important factors were recognized: the mother's educational level across the entire database and in the Nord region for stunting and underweight, and the place of residence in the Nord region for underweight. However, this is consistent with several previous studies [1,12,13,2,41,46] and suggests that, although modifiable, these same factors persist over the years in African countries. Such persistence could indicate that efforts

to combat undernutrition are either insufficiently focused on modifiable determining factors, such as mother's education, or are inappropriate. Further studies should provide more insight into this issue, and appropriate programs focusing specifically on these modifiable factors should be implemented to improve the nutritional status of children under five years of age.

This study is not without limitations. The main limitation of this study was its secondary level. Data were selected from various primary databases based on criteria such as completeness and plausibility of the anthropometric parameters of children under 5 years old, which reduced the sample size compared to other studies that used the same survey databases in Burkina Faso [22]. Moreover, this selection process did not account for the hierarchical structure of the data during extraction. Furthermore, the sample sizes in the health regions and districts were not evenly distributed. However, we conducted an analysis of the hierarchical structure by examining the evolution of undernutrition in children across health regions and districts.

## Conclusion

Based on the evidence, our country has evolved over the decades through various health strategies aimed at improving the health system, healthcare, and population health indicators. An increasing number of integrated packages now include health, nutrition, agriculture, water, hygiene, sanitation, links to healthcare, women's empowerment, income generation, advocacy, and more. The present study aimed to examine the evolution of the nutritional status of children under 5 years of age between 2014 and 2017, as well as the factors associated with undernutrition, based on surveys conducted for various primary purposes in Burkina Faso. It allowed us to demonstrate how the situation of children's undernutrition evolved between 2014 and 2017. From higher prevalence, it reached lower levels of stunting, wasting, and underweight, but without completely eliminating them. This encouraging evolution should not lead to complacency in the fight against malnutrition in children under five years old, but rather intensify efforts to achieve all objectives of ending undernutrition. However, this encouraging trend was observed before the current insecurity in the country and its severe consequences. Moreover, our study highlighted the persistence of modifiable factors associated with undernutrition in children, along with the emergence of new ones. Therefore, to close the undernutrition gap, long-term, appropriate strategies are required to target multi-sectoral solutions, along with rigorous evaluation to optimize results for global and sustainable reduction, regardless of the context.

## Supporting information

**S1 Fig. Trends in the mean Z-scores of children by age group between 2014 and 2017 in Burkina Faso.**
(TIF)

**S2 Fig. Trends in the Mean Z-Scores of Children by Region Between 2014 and 2017 in Burkina Faso.** The base layer of this map comes from GADM (https://gadm.org/download_country.html), public domain. This resource is compatible with the CC-BY 4.0 license:https://gadm.org/license.html.
(TIF)

**S1 Data. Study database from the surveys of the impact evaluation of the PBF program in Burkina Faso.**
(DTA)

## Acknowledgments

The authors would like to acknowledge the entire staff of the Centre de Calcul du Centre MURAZ for their contributions to this study. We would like to express our gratitude and appreciation, especially to André SOME, Eric DABONE, Mostafa BERDII, Djamal TOE, Seydou DRABO, and Dr. Alamissa SOULAMA for their contributions to this work.

## Author contributions

**Conceptualization:** Tarwendpanga Bernadette PICBOUGOUM.

**Data curation:** Tarwendpanga Bernadette PICBOUGOUM.

**Formal analysis:** Tarwendpanga Bernadette PICBOUGOUM.

**Methodology:** Tarwendpanga Bernadette PICBOUGOUM, M.A. Serge SOMDA, Annie ROBERT.

**Writing – original draft:** Tarwendpanga Bernadette PICBOUGOUM.

**Writing – review & editing:** Tarwendpanga Bernadette PICBOUGOUM, M.A. Serge SOMDA, Isidore T. TRAORE, Julia LOHMANN, Manuela DE ALLEGRI, Hervé HIEN, Nicolas MEDA, Annie ROBERT.

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
