## [Decision Letter · Decision Letter 0]

14 Apr 2025

PGPH-D-25-00211

Evolution of the nutritional status and factors associated with undernutrition in children under five of age between 2014 and 2017 in 24 health districts of Burkina Faso

Dear Dr. PICBOUGOUM,

Thank you for submitting your manuscript to PLOS Global Public Health. After careful consideration, we feel that it has merit but does not fully meet PLOS Global Public Health’s publication criteria as it currently stands. Therefore, we invite you to submit a revised version of the manuscript that addresses the points raised during the review process.

We look forward to receiving your revised manuscript.

Kind regards,

Hasanain Faisal Ghazi, phd

Academic Editor

Journal Requirements:

1.    We have amended your Competing Interest statement to comply with journal style. We kindly ask that you double check the statement and let us know if anything is incorrect.

2.    We do not publish any copyright or trademark symbols that usually accompany proprietary names, eg (R), (C), or TM  (e.g. next to drug or reagent names). Please remove all instances of trademark/copyright symbols throughout the text, including ® on page 13.

Additional Editor Comments (if provided):

please respond to comments

Reviewers' comments:

Reviewer's Responses to Questions

**Comments to the Author**

1. Does this manuscript meet PLOS Global Public Health’s publication criteria ? Is the manuscript technically sound, and do the data support the conclusions? The manuscript must describe methodologically and ethically rigorous research with conclusions that are appropriately drawn based on the data presented.

Reviewer #1: Yes

Reviewer #2: Yes

2. Has the statistical analysis been performed appropriately and rigorously?

Reviewer #1: No

Reviewer #2: Yes

3. Have the authors made all data underlying the findings in their manuscript fully available (please refer to the Data Availability Statement at the start of the manuscript PDF file)?

Reviewer #1: No

Reviewer #2: Yes

4. Is the manuscript presented in an intelligible fashion and written in standard English?

Reviewer #1: Yes

Reviewer #2: Yes

5. Review Comments to the Author

Reviewer #1: The research work on Evolution of the nutritional status and factors associated 2 with undernutrition in children under five of age between 2014 and 3 2017 in 24 health districts of Burkina Faso has been reviewed and the following comments are as follows

1.Abstract and Introduction Chapter:

•Mention the nutritional status prevailing in the study groups as per WHO data

•Its mentioned thst “From 2014 to 2017, the proportion 38 of severe stunting and severe underweight decreased” mention the reason for it.

•The statement “to assess the evolution of the nutritional status of children 94 under 5 years of age, as well as the factors associated with undernutrition, between 2014 and 95 2017 in Burkina Faso” what do you mean by evolution of nutritional status? Specify the exact aim/objective of the research work.

•Address the Problem statement, research gap, existing scenario clearly in the introduction

2. Materials and Methods

Its rework of the data collected already. what is the method used for analysis? The method used in previous work is different from this? If so, then what is the gap identified?

In Analysis – What is the significance of Z- score? The analysis conducted was based on the significance of what?

How do you conclude the abnormalities are reduced? How do you compare your work with the latest results?

Conclusion:

What are all the modifiable factors identified?

Comment on the reopening this research study now. What significance you have depicted out of your research?

Reviewer #2: Brilliant manuscript, good writeup and rigorous statistical analysis.However I have few concerns.

Line 110-111- Kindly clarify what is meant by pre and post implementation assessment, was any intervention carried out? Clearly state so if any.

Line 126- Clearly state how malaria infection and anemia was diagnosed among those participants at the household.

Line 165- The acronym (LR) should immediately follow Logistic regression so that subsequently one can follow when only the acronym is used as in Line 168.

Once these minor concerns are rectified, this is a good article and I recommend it for publishing. Thank you

6. PLOS authors have the option to publish the peer review history of their article (what does this mean? ). If published, this will include your full peer review and any attached files.

**Do you want your identity to be public for this peer review?** For information about this choice, including consent withdrawal, please see our Privacy Policy .

Reviewer #1: **Yes: ** Hemalatha RJ

Reviewer #2: No

---

## [Editor Report · Decision Letter 1]

13 May 2025

Evolution of the nutritional status and factors associated with undernutrition in children under five of age between 2014 and 2017 in 24 health districts of Burkina Faso

PGPH-D-25-00211R1

Dear Dr PICBOUGOUM,

We are pleased to inform you that your manuscript 'Evolution of the nutritional status and factors associated with undernutrition in children under five of age between 2014 and 2017 in 24 health districts of Burkina Faso' has been provisionally accepted for publication in PLOS Global Public Health.

Best regards,

Hasanain Faisal Ghazi, phd

Academic Editor